# Nanocapsules Comprised of Purified Protein: Construction and Applications in Vaccine Research

**DOI:** 10.3390/vaccines12040410

**Published:** 2024-04-12

**Authors:** Ivana Skakic, Aya C. Taki, Jasmine E. Francis, Chaitali Dekiwadia, Thi Thu Hao Van, Carina C. D. Joe, Tram Phan, George Lovrecz, Paul R. Gorry, Paul A. Ramsland, Anna K. Walduck, Peter M. Smooker

**Affiliations:** 1School of Science, RMIT University, 264 Plenty Road, Bundoora, VIC 3083, Australia; ivana.skakic@rmit.edu.au (I.S.); j.ericafrancis@gmail.com (J.E.F.); thithuhao.van@rmit.edu.au (T.T.H.V.); paul.ramsland@rmit.edu.au (P.A.R.); 2Melbourne Veterinary School, Faculty of Science, The University of Melbourne, Parkville, VIC 3010, Australia; aya.taki@unimelb.edu.au; 3RMIT Microscopy & Microanalysis Facility, School of Science, RMIT University, Melbourne, VIC 3001, Australia; chaitali.dekiwadia@rmit.edu.au; 4CSIRO Manufacturing, Clayton, VIC 3169, Australia; carina.joe@outlook.com (C.C.D.J.); tram.phan@csiro.au (T.P.); glovrecz@gmail.com (G.L.); 5Department of Microbiology and Immunology, University of Melbourne, Peter Doherty Institute of Infection and Immunity, Parkville, VIC 3010, Australia; prgorry@unimelb.edu.au; 6Department of Immunology, Monash University, Melbourne, VIC 3004, Australia; 7Department of Surgery, Austin Health, The University of Melbourne, Heidelberg, VIC 3084, Australia

**Keywords:** nanoparticles, vaccines, influenza virus, *Helicobacter pylori*, HIV

## Abstract

Nanoparticles show great promise as a platform for developing vaccines for the prevention of infectious disease. We have been investigating a method whereby nanocapsules can be formulated from protein, such that the final capsules contain only the cross-linked protein itself. Such nanocapsules are made using a silica templating system and can be customised in terms of size and porosity. Here we compare the construction and characteristics of nanocapsules from four different proteins: one a model protein (ovalbumin) and three from infectious disease pathogens, namely the influenza virus, *Helicobacter pylori* and HIV. Two of the nanocapsules were assessed further. We confirm that nanocapsules constructed from the urease A subunit of *H. pylori* can reduce subsequent infection in a vaccinated mouse model. Further, we show that capsules constructed from the HIV gp120 protein can be taken up by dendritic cells in tissue culture and can be recognised by antibodies raised against the virus. These results point to the utility of this method in constructing protein-only nanocapsules from proteins of varying sizes and isoelectric points.

## 1. Introduction

The emergence of SARS-CoV-2 and the resultant COVID-19 pandemic taught us a great deal, with a major lesson being the need to be able to quickly develop vaccines using a variety of different technologies, some of which had not been used in human vaccines previously. In preparation for the emergence of new pathogens, which almost certainly will happen, it is vital to keep developing a variety of platforms for vaccines. To this end, our laboratory has been testing a variety of different platforms over the years, from naked DNA and recombinant protein vaccines to DNA vaccines delivered using solid-lipid nanoparticles and nanoparticles comprised of pure protein. In this manuscript we will focus on the latter, illustrating some of the particles we have designed and developed and describing experimental uses for two of the particles which demonstrate their utility.

The concept of using nanoparticles as a vaccine delivery system has been around for at least two decades and revolves around the concept that nanoparticles have some advantages over subunit vaccines, be they DNA or protein. For examples of recent reviews on nanoparticle vaccines we direct the reader to Wang et al. (2023) [1] and Zhou et al. (2023) [2]. A major feature of nanoparticle vaccines is their size which, as their name suggests, is less than one micron. One major advantage of this is that they can be formulated to be the ideal size for uptake by immune cells such as dendritic cells (DCs) and macrophages, thus delivering their cargo into such cells for subsequent antigen processing and presentation to T cells. Equally, the nanoparticles can be recognised by B cells for subsequent antibody induction. Once a nanoparticle has been captured by, e.g., a DC after vaccination, the cell can instigate migration to the draining lymph node to initiate the immune response [3]. It has been shown that nanoparticles of sufficiently small size (less than around 200 nm) can also freely drain through the lymph and transit the lymphatic capillary. The smaller the particle the more efficient this method of lymph node targeting appears [4]. This can result in the efficient induction of immune responses in the lymph node.

No introduction is complete without mention of the nanoparticles used to deliver mRNA including, but not limited to, lipid nanoparticles [5]. These are undoubtedly the best-known vaccine nanoparticles (although perhaps the general public do not recognise them as that!). These particles have a size range reported as around 50–100 nm, a very common size for nanoparticles. The hydrodynamic size range of the Pfizer-BioNTech BNT162b2 COVID-19 vaccine has been measured as just over an average of 90 nm, for example [6]. It is likely that there will be a major push in the use of various types of lipid nanoparticles to deliver coding regions not only in the form of mRNA but also DNA [7,8,9].

Some of the concerns with the use of nanoparticle vaccines have revolved around the potential toxicity of the materials used [10]. In this report, we detail the construction and evaluation of nanoparticles comprised of pure crosslinked protein, which largely alleviates such concerns. The rationale for this work was to use the “templating” approach first described as a vehicle for the delivery of a cargo in drug delivery [11]. They were the first to use solid core/mesoporous shell (SC/MS) silica particle templates in this way, using a variety of polymers to form the capsules. We reasoned that by using protein as the polymer we could make potentially immunogenic particles of a defined, monodispersed size (as the size is dependent on that of the template). These are not designed to deliver a cargo (although they potentially could). Here we demonstrate the successful construction of four nanoparticles (termed nanocapsules) from different proteins, highlighting the broad applicability of the technique. Lastly, we evaluated two of these in experimental systems. It should be noted that, of the four nanocapsules described here, two are for illustrative purposes and two for experimental analysis. The aim was to demonstrate that nanocapsules can be synthesised from a range of proteins. Ovalbumin nanocapsules have been described by us previously [12], while C27 nanocapsules were constructed as we wanted to see if we could make stable nanocapsules from a protein of relatively low molecular mass. This was achieved. They were not characterised further. Nanocapsules constructed from urease A (UreA) from *Helicobacter pylori* were tested in a mouse model of infection, while those constructed from the gp120 protein from the HIV virus were tested for immunogenicity in vitro.

## 2. Materials and Methods

### 2.1. Preparation of Silica Templates

The silica templates into which the target proteins were infused were constructed as reported in our recent publications [12,13,14]. The majority of the particles constructed were solid core (SC)/mesoporous shell (MS) particles of approximate diameter of 500 nm. A very useful advantage of these methods is that it is possible to construct fully porous particles of a smaller diameter, and we have constructed such MS particles of approximately 50 nm diameter.

For the SC/MS particles, the solid core was prepared by the hydrolysis and condensation of tetraethoxysilane (TEOS, Sigma-Aldrich, Darmstadt, Germany) in ethanol, MQH_2_O (Merck Millipore, Burlington, MA, USA) and ammonium hydroxide (Merck, Darmstadt, Germany). Following this, a sol–gel coating of a mesoporous shell (MS) (for the 500 nm templates) was overlayed onto the pre-formed solid core (SC) by adding TEOS (Sigma-Aldrich, Darmstadt, Germany) and TMS (Sigma-Aldrich, Darmstadt, Germany) to a solution containing SC particles over a 20 min period while stirring. The particles were incubated for 2.5 h, enabling the formation of the MS. The SC/MS templates were washed in ethanol and collected by centrifugation at 5000× *g* and air-washed in distilled H_2_O and dried overnight at room temperature. The porogen TMS was removed through calcination by heating at 550 °C for 6 h.

Fully porous MS particles (for 50 nm templates) were prepared by combining MQH_2_O, ethanol and 25% cetyltrimethylammonium chloride (CTAC, Sigma-Aldrich, Darmstadt, Germany) while stirring for 10 min at room temperature. Subsequently triethanolamine (TEA, Sigma-Aldrich, Darmstadt, Germany) was added slowly and stirred until dissolved. Lastly, 20 mL of this solution was heated to 60 °C for 2 h without stirring, and subsequently 1.454 mL of TEOS was added dropwise while stirring, then left to cool. The templates were collected by centrifugation at 47,800× *g*, washed in distilled H_2_O and dried overnight at room temperature. The templates were calcinated by heating at 550 °C for 6 h. The calcined templates were then stored dry until use. Readers are directed to Skakic et al. (2022) for the full protocol of template production [14].

### 2.2. Preparation of Proteins

For the construction of the model protein ovalbumin capsules, a stock of 100 mg/mL albumin from chicken egg white, Grade V (Sigma-Aldrich, Darmstadt, Germany) in MQH_2_O was used. Each of the other three proteins tested was made as a recombinant protein, and coding sequences were cloned into expression vectors as follows:
H1N1 influenza haemagglutinin fragment c27: pET45b: c27. A gene segment encoding amino acids 301–404 of the Haemagglutinin gene from A/Puerto Rico/8-SV8/1934(H1N1) (Accession # AEX92912.1) was inserted in frame with the leader sequence encoded by the pET-45b vector, using standard cloning procedures. The encoded protein had the following sequence, with the vector encoded sequence underlined:MAHHHHHHVGTGSNDDDDKSPDPGAINSSLPYQNIHPVTIGECPKYVRSAKLRMVTGLRNIPSIQSRGLFGAIAGFIEGGWTGMIDGWYGYHHQNEQGSGYAADQKSTQNAINGITNKVNTVIEKMN*H. pylori* urease A subunit: pRSETa-Ure A. The construction and expression of this protein has been previously described [13].


The above two proteins were expressed in the *E. coli* BL21-DE3 strain and purified by standard immobilised metal-affinity chromatography (IMAC).

3.The HIV gp120 protein sequence was taken from Uniprot (ID: P04578) and was reverse translated and codon optimised for mammalian expression using the IDT codon optimisation tool (Integrated DNA Technologies, Coralville, IA, USA). For protein detection and purification purposes, a 6xHis-tag was added to the N-terminus of the gp120 sequence, and the native signal peptide was added to the N-terminus of the 6xHis-tag. The gp120 protein was 1554 aa in length (inclusive of the 6xHis-tag). The 6539 bp codon optimised construct was purchased in mammalian expression vector pcDNA3.1(-)zeo (Genscript, Piscataway, NJ, USA) and delivered as lyophilised plasmid. The mammalian cell line HEK293 was used for recombinant protein expression using the Freestyle293 Expression System (ThermoFisher, Waltham, MA, USA), as recommended by the manufacturer.

### 2.3. Nanocapsule Synthesis and Characterisation Protocols

The nanocapsules were prepared by the incubation of silica templates with purified protein at a 3:1 (*w*/*w*) ratio in PBS (sodium chloride 0.8%, potassium chloride 0.02%, disodium hydrogen phosphate 0.115%, potassium dihydrogen phosphate 0.02%, pH 7.4) overnight at 4 °C on an end-over-end suspension mixer. Either ~500 nm SC/MS or ~50 nm MS templates were used. After infiltration, excess protein was removed by PBS washes. The proteins were cross-linked using 5% (*w*/*v*) glutaraldehyde (GA, Sigma-Aldrich, Darmstadt, Germany) for 2 h at 4 °C on an end-over-end suspension mixer, after which excess GA was removed by washing in PBS. Finally, the silica templates were removed by 80% (*v*/*v*) of 2 M hydrofluoric acid treatment. The resulting protein-based nanocapsules were washed, resuspended in sterile PBS and stored at 4 °C until use.

### 2.4. Vaccine Trial: H. pylori UreA Nanocapsules

Materials and methods used for the vaccine trial were as previously reported, with minor modifications [13]. All animal experiments were undertaken with approval from the RMIT Animal Ethics Committee under the Animal Ethics Committee: approval number AEC1212. All experiments and procedures were undertaken following approved animal handling and experimental protocols.

Six-week-old female *Helicobacter*-free C57BL/6 mice were purchased from the Animal Resource Centre, Canningvale, Western Australia. The mice were fed a sterile standard diet, had access to water ad libitum and were housed under specific pathogen-free conditions. Groups of eight mice were immunised with PBS (sham) or nanocapsules twice via the intraperitoneal route, according to the scheme shown in Table 1.

The nanocapsules were formulated with or without TiterMax adjuvant. For the third group, the nanocapsules were emulsified in TitreMax Gold adjuvant (Sigma-Aldrich, Darmstadt, Germany) in a 1:1 water-in-oil emulsion, as recommended by the manufacturer. A dose of 50 µL of each emulsion was injected.

### 2.5. Tissue Analyses and Bacterial Burden Post-Challenge

Three weeks after the second vaccination, the mice were challenged by oral gavage [15]. The dose was 0.1 mL containing 10^7^ CFU *H. pylori* Sydney strain SS1 [16]. Three weeks after the challenge, the mice were killed, and tissues were removed and assessed for gastric bacterial load and immune cell responses as described previously [13]. Mesenteric lymph nodes (MLN) were also assessed for cellular responses.

DNA was extracted from stomach homogenates (half stomach from five mice per group) as described in Skakic et al. (2023) [13]. The burden of *H. pylori* was determined using a SYBR green qPCR specific for the *H. pylori* 16S RNA and quantified against *H. pylori* DNA standards ranging from 0.0001 to 100 ng, as described in Tan et al. (2008) [17], using Bioline SYBR Green No-ROX Mastermix (Meridian Bioscience, Lukenwalde, Germany), containing 300 ng DNA, 0.4 μM forward and reverse primers (Table 2). The conditions for qPCR were as follows: 95 °C, 10 min, 45 cycles, 95 °C 10 s, 60 °C. Melt curve analysis was used to confirm the specificity of the amplification products. The DNA extracted from PBS naïve control stomachs was used to set the baseline above which a signal threshold was applied. The results were expressed at *H. pylori* genomes/g tissue.

The complete stomach of 3–5 mice/group was used for flow cytometry analysis. Cells were isolated and inflammatory infiltrates in the stomach and MLN were analysed by flow cytometry as previously described [13]. The following targets and antibodies were used for surface staining: CD3 (145-2C11), CD45 (30-F11), CD4 (GK1.5), (all antibodies sourced from BioLegend, San Diego, CA, USA), followed by intracellular staining for IFNγ (XMG1.2) using the Fix/Perm kit (BD Biosciences, Franklin Lakes, NJ, USA).

### 2.6. Evaluation of Dose-Dependent Cytotoxicity of gp120 Nanocapsules

The cytotoxicity of gp120 nanocapsules on DC2.4 cells was evaluated at a range of concentrations, using a PrestoBlue^®^ Cell Viability reagent (ThermoFisher, Waltham, MA, USA). DC2.4 cells were seeded in a 96 well plate at a density of 5 × 10^3^ cells per well in 100 µL of RPMI media supplemented with 10% heat inactivated FBS and 1% Penicillin Streptomycin antibiotic and incubated at 37 °C in a 5% CO_2_ humidified atmosphere overnight. Following overnight incubation, the media was replaced by 100 μL of fresh complete RPMI containing a final concentration of 5, 10, 25, 50 or 100 μg/mL gp120 nanocapsules and incubated for 24, 48 or 72 h.

At each time-point, 10 µL of PrestoBlue^®^ Cell Viability reagent was added to cells and incubated at 37 °C in a 5% CO_2_ humidified atmosphere for 60 min in the dark. Absorbance was read at 570 nm with a 600 nm reference wavelength for normalisation using the POLARSTAR Omega microplate reader (BMG Labtech, Ortenberg, Germany). Cells treated with media alone were treated as the negative control and assumed to represent 100% viability. Cells treated with 0.01% Triton-X were treated as the positive control and assumed to represent complete cell death. Percentage of cell viability was calculated by setting the negative control to 100%. Statistical significance of time-dependant versus dose-dependent toxicity was calculated by two-way ANOVA (significance threshold = 0.05).

### 2.7. Characterisation of gp120 Nanocapsule Uptake by Fluorescence Microscopy

To observe cellular uptake of gp120 nanocapsules, DC2.4 cells were exposed to either SC/MS gp120 nanocapsules or MS gp120 nanocapsules for 6 h and imaged by fluorescence microscopy. The cells were seeded onto 25 mm glass coverslips in a 6-well plate, at a density of 2 × 10^5^ cells per well, and incubated at 37 °C in 5% CO_2_ humidified atmosphere for 24 h. The cells were treated with 100 μg/mL of either SC/MS gp120 nanocapsules or MS gp120 nanocapsules in RPMI 1640 GlutaMAX™ medium (ThermoFisher, Waltham, MA, USA) with 10% heat-inactivated FBS and incubated at 37 °C for 6 h to allow for cellular uptake of the nanocapsules. Cells kept under identical conditions, without exposure to nanocapsules, served as a negative control.

Following incubation, free nanocapsules were removed by gentle washing with cold PBS three times. The cells were then permeabilized by adding 0.1% Triton X-100 for 5 min. Following this, the cells were fixed onto coverslips by adding 4% PFA for 15 min at room temperature, followed by washing with PBS three times. Nuclear staining with DAPI was undertaken at room temperature for 15 min, followed by washing with PBS three times. Following this, the cells were stained with Alexa Fluor 488 Phalloidin (ThermoFisher, Waltham, MA, USA), which binds to actin filaments of the cell and allows for visualisation of the cell structure. The coverslips were mounted and sealed onto a glass slide using gold antifade reagent (ThermoFisher, Waltham, MA, USA). The nanocapsules were autofluorescent in the red channel at 550 nm due to the GA cross-linking, and consequently no fluorescent labelling was required. Untreated cells served as a negative control. Cellular imaging was performed using a Leica DM2500 Epifluorescence microscope (Leica, Wetzlar, Germany), and image capture and analysis were performed using Leica Application Suite (LAS, version 4.10.0) software.

### 2.8. Evaluation of Epitope Availability of gp120 Nanocapsules via Indirect Enzyme-Linked Immunosorbent Assay (ELISA)

Indirect ELISA was used to measure epitope availability on gp120 nanocapsules, and all samples were tested in duplicate, and each assay was repeated three times. Ninety-six-well flat-bottomed sterile immunosorbent assay plates were coated with 100 µL/well of 5 µg/mL purified soluble gp120 protein, SC/MS gp120 nanocapsules, or MS gp120 nanocapsules in a coating buffer and incubated overnight at 4 °C. Of the soluble gp120 protein control wells, one set of duplicate wells was used as a positive control (termed ‘soluble gp120’) and one set of duplicates was used as a glutaraldehyde control (termed ‘soluble gp120 + GA) to account for autofluorescence from GA. The soluble gp120 + GA control was cross-linked with 5% (*w*/*v*) GA for 2 h at 4 °C, following overnight coating. Primary and secondary antibodies were incubated, and the reaction was visualised using standard procedures [13].

The primary antibodies used in this study were PG16, targeting an epitope in the V1/V2 loop domain [19]; NIH45-46, targeting the CD4 binding site [20]; and 2G12, targeting the conserved glycan region of GP120 [21].

## 3. Results

### 3.1. Construction and Characterization of the Nanocapsules

A schematic representation of the construction of the nanocapsules is given in Figure 1. The procedure is quite straightforward: with the generation of silica templates, infiltration and cross-linking of the polymer (in our case, protein) and then the removal of the sacrificial silica template.

In constructing the nanocapsules, firstly two different-sized templated were synthesized. The first was a SC/MS template approximately 500 nm in diameter, and the second a fully MS template approximately 50 nm in diameter. Representative electron micrographs of the silica templates are shown in Figure 2.

Following template synthesis, protein was infiltrated into the pores and subsequently cross-linked to form a stable protein structure. The silica template was then removed, leaving a hollow protein shell (large nanocapsules) or a protein particle (small nanocapsules). Representative micrographs of the nanocapsules are depicted in Figure 3.

In addition to characterization by TEM, the nanocapsules were analysed by DLS to estimate the hydrodynamic size and (for some) the zeta potential (Table 3). For each of the nanocapsules, the hydrodynamic size measurements correlated well with the diameters estimated by TEM, thereby indicating minimal aggregation in the solution. The zeta potential of each of the particles measured was similar, from −14 to −17 mV. While these values do not necessarily indicate high stability, coupled with the hydrodynamic size estimates, it appears that the propensity for the nanocapsules to the aggregate is minimal.

### 3.2. Vaccine Trial

#### 3.2.1. *H. pylori* Colonisation

*H. pylori* burden was determined by qPCR specific for the 16sRNA gene for 3–8 mice per group, as shown in Figure 4. The numbers of *H. pylori* genomes were calculated from a standard curve as previously described [18]; values were Log10 transformed. Statistical significance was determined using the Mann–Whitney U-test. The limit of detection for the assay was determined to be 10^2^/g. Mice vaccinated with UreA nanocapsules (nc) had on average a 1 log reduction in colonization 21 days after challenge compared to the sham-vaccinated, challenged controls (*p* = 0.033, 2-tailed test). Mice that received UreA nc with an adjuvant also displayed significantly reduced burdens (*p* = 0.0485, 2-tailed test). As expected, *H. pylori* was not detected in non-challenged controls.

#### 3.2.2. Characterisation of Immune Cell Populations

Infiltrating immune cell populations in mouse stomach tissue were stained for relevant surface markers and analysed by flow cytometry, as shown in Figure 5. Statistical significance was determined by a Mann–Whitney U-test in comparison to the sham-vaccinated, challenged controls. Data from n = 3–5 individuals per group are shown. Overall, low cell numbers were isolated from individual mice, consistent with our previous reports. Mice that received UreA nanocapsules with adjuvant had mildly increased levels of infiltrating CD4^+^ cells in the stomach, although considerable variation was observed (Figure 5A). This correlated to raised numbers of IFNγ^+^ CD4^+^ cells in UreA nanocapsule and UreA nanocapsule with adjuvant groups (Figure 5B). The mean number of cells was higher in the nanocapsules plus adjuvant group, indicating that the adjuvant may be required for maximal recruitment of cells. No significant effect was detected in the MLN, however (Figure 5C).

### 3.3. Evaluation of gp120 Nanocapsules

#### 3.3.1. Dose-Dependent Cytotoxicity

The cytotoxicity of SC/MS and MS gp120 nanocapsules towards DC2.4 cells was evaluated at a range of concentrations by PrestoBlue^®^ Cell Viability assay. DC2.4 cells were treated for up to 72 h with either SC/MS gp120 nanocapsules or MS gp120 nanocapsules, at which point viability was compared to the control of cells only in media. Figure 6A shows the viability of cells treated with SC/MS gp120 nanocapsules over a 72 h period, with viability assessed every 24 h. Cell viability was maintained above 73% for all concentrations across all time points. Figure 6B shows the viability of cells treated with MS gp120 nanocapsules over a 72 h period, with viability also assessed every 24 h. As with the SC/MS gp120 nanocapsules, the MS nanocapsules showed limited signs of toxicity toward the DC2.4 cells, with viability maintained above 70% for all concentrations and across all time points. These results indicate that, overall, gp120 nanocapsules displayed some cytotoxicity at high concentrations. Notably, a downward trend was observed in SC/MS gp120 nanocapsules at 72 h in the 50 µg/mL and 100 µg/mL samples, and a similar decrease was seen in the MS gp120 nanocapsules in the 25 µg/mL, 50 µg/mL, and 100 µg/mL. All viability levels were maintained above 70%, and cytotoxic effects were found to be both time- and dose-dependent.

#### 3.3.2. Characterisation of gp120 Nanocapsule Uptake by Fluorescence Microscopy

To observe the cellular uptake of gp120-based nanocapsules, DC2.4 cells were exposed to either SC/MS gp120 or MS gp120 nanocapsules for 6 h and imaged by fluorescence microscopy. The cells not treated with nanocapsules served as a negative control. Cellular imaging was performed using a Leica DM2500 Epifluorescence microscope, and image capture and analysis were performed using the Leica Applications Suite (LAS, version 4.10.0) software. The use of GA to cross-link gp120 in this study allowed use of its autofluorescent properties in cellular imaging, without the requirement of labelling the nanocapsules with a fluorescent tag.

Fluorescence micrographs of the control cells (untreated) or cells treated with either SC/MS or MS gp120-based nanocapsules are shown in Figure 7. Images acquired in the DAPI channel show the DAPI-stained cell nuclei, the actin skeleton of the cell is indicated in the green channel, and images in the red channel show cells which have taken up fluorescent nanocapsules (free nanocapsules had been removed by washing). It is evident that there is successful in vitro uptake of gp120 nanocapsules within 6 h of exposure to DC2.4 cells.

#### 3.3.3. Evaluation of Epitope Availability of gp120 Nanocapsules via Indirect ELISA

To determine availability of known gp120 epitopes in gp120 nanocapsules, nanocapsules were probed with anti-gp120 antibodies to determine titre as a measure of epitope availability. Titre levels were measured by ELISA and compared to soluble gp120, and gp120 treated with GA.

Epitope binding by anti-2G12 antibody is shown in Figure 8A. Soluble gp120 indicated highest titre levels, with a mean titre of 2.9 × 10^6^, followed by the gp120 + GA control with a mean titre of 1.5 × 10^5^. The SC/MS nanocapsules indicated a mean titre of 6.7 × 10^4^, and the MS nanocapsules showed a significant decrease in epitope availability compared to the soluble gp120 (*p* ≤ 0.01) with mean titre of 2.3 × 10^4^.

Epitope availability indicated by probing with NIH45-46 is shown in Figure 8B and shows that the soluble gp120 control had the highest epitope availability followed by the gp120 + GA control, with mean titres of 6.8 × 10^6^ and 7.5 × 10^5^, respectively. The SC/MS nanocapsules and MS nanocapsules showed a significant decrease in epitope availability (*p* ≤ 0.05), compared to soluble gp120, with mean titres of 1.2 × 10^5^ and 8.7 × 10^4^, respectively.

PG16 epitope analysis indicates that soluble gp120 has the highest level of epitope availability for recognition, with a mean titre of 3.7 × 10^6^ (Figure 8C). The epitope availability was also high in the SC/MS nanocapsules and gp120 + GA control, with mean titres of 1.92 × 10^6^ and 1.78 × 10^6^, respectively. The lowest levels of epitope availability for PG16 were seen in the MS nanocapsules, with a mean titre of 2.4 × 10^5^. There was no significant reduction in epitope availability in the gp120 + GA, SC/MS nanocapsules, or MS nanocapsules compared to the soluble gp120 protein.

The soluble gp120 protein titres were demonstrably highest for all three antibodies assessed, compared to the soluble gp120 cross-linked with GA, the SC/MS nanocapsules, or MS nanocapsules. This data demonstrates that despite a significant reduction in the availability of epitopes after nanocapsule formation (as might be expected), an antibody is able to recognise the nanocapsules.

## 4. Discussion

This manuscript details the use of a “templating” approach to produce nanocapsules that contain only cross-linked protein. As demonstrated, such nanocapsules can be constructed from either native or recombinant protein of varying molecular mass and isoelectric point and from a range of species. Theoretically, any protein could be employed to produce such particles. Further, by adjusting the chemistry in the manufacture of the silica templates, capsules of varying sizes can be created, with either a porous shell or a completely porous particle. Those formed based on SC/MS templates will be hollow particles, with the latter (MS) more rigid as the protein is infused throughout the template prior to its removal.

These nanocapsules are therefore another type of potential vehicle for the development of vaccines. As has been evidenced by many studies, there are “horses for courses”; that is, a particular platform may be more applicable than others against a pathogen. This is usually dependent on the delivery mechanism and the desired immune responses [22,23]. This will be discussed further below.

In terms of the synthesis of the nanocapsules, it can be seen that irrespective of the nature of the protein, the capsules were quite uniform (Figure 3). This is to be expected, as the template determines the size. Each of the nanocapsules derived from the SC/MS templates are around the designed 500 nm in diameter, and the two MS-derived capsules are approximately 50 nm. Although we have no direct evidence, it will be possible to further vary the size of the nanocapsules by modifying the chemistry used in template formation. In particular, the solid core size can be varied. It has previously been shown that reaction temperature and ammonia concentration influence template sizing [24]. In that report, silica cores of 170, 290, and 490 nm diameters were formulated. Once an MS shell is deposited, nanocapsules of a similar size could be constructed. This approach can therefore facilitate the production of protein nanocapsules of a desired size, which can dictate the pathway by which the antigen is taken up and processed [25]. Protein characteristics such as molecular mass and isoelectric point may influence the efficiency of template formation (for example, by defining the rate of template pore infiltration) but were not a barrier to the formation of any of the four proteins evaluated here.

Two of the nanocapsules were evaluated further. The first, UreA, was evaluated in a mouse model of *H. pylori* infection. We have previously demonstrated that large UreA nanocapsules, but not smaller capsules, can induce protection in this model [13]. Here we have confirmed protection by these large capsules. Thus, the 500 nm nanocapsules comprised of the urease A subunit protein were able to induce a protective response in mice, as shown in Figure 4. This compares to a similar experiment using the same challenge system, where mice were vaccinated with a DNA vaccine encoding the same antigen, UreA. No decrease in colonization was seen [8]. As alluded to above, different vaccine platforms may be suited to particular host–pathogen combinations, in terms of vaccinating to induce protective responses.

An analysis of the immune responses induced in this study indicates the ability of the nanocapsules to initiate the migration of CD4^+^ T cells into the gastric tissue. As it has been demonstrated that such cells are critical for protection in this mouse model [26], it may be that such nanocapsules can be further evaluated as a vaccine against *H. pylori* infection. Further work should be aimed at determining the specificity of the infiltrating T cells.

In the second evaluation, the availability of known gp120 epitopes was tested using specific monoclonal antibodies. Figure 9 shows a schematic representation of the targets of these antibodies.

The availability of key conserved epitopes in vaccine design against HIV is critical to trigger T-cell and B-cell reactions and induce robust and appropriate immune responses [27,28]. The gp120 protein has a range of known epitopes that are of interest as vaccine targets. To determine whether the use of gp120 in nanocapsule form maintained the availability of known epitopes, capsules were compared to free gp120 protein and gp120 treated with GA without infiltration. The results indicated that in two of three epitopes, availability was significantly reduced in both SC/MS and MS nanocapsules compared to soluble gp120. Availability of the PG16 epitope was not significantly reduced. The reduction in availability is not unexpected, as some B-cell epitopes will no longer be on the surface, thereby reducing availability per unit mass of nanocapsule compared to free protein.

The 2G12 antibody recognises glycans contained within the highly conserved glycan cluster of the gp120 outer domain, binding to a carbohydrate-dependant epitope [29]. Epitope availability analysis indicated no significant reduction in 2G12 binding in the gp120 + GA and SC/MS nanocapsules groups, compared to soluble gp120, and similar results were observed in the gp120 + GA group for NIH45-46 and PG16. However, despite lack of statistically significant decreases, it is notable that treatment with glutaraldehyde did indicate a slight reduction in epitope availability in 2G12 and NIH45-46, indicating that crosslinking alone (without infiltration into nanoparticle templates) reduces available epitopes to some degree.

A significant decrease in epitope recognition was observed in SC/MS nanocapsules and MS nanocapsules with NIH45-46, an antibody which recognises glycans surrounding the CD4 binding site. This reduction in epitope recognition may be due to the location and exposure of the binding site; the CD4 binding site is contained within a depression which is formed at the interface of the outer domain and inner domain of the gp120 subunit [30]. As such, once the protein is infiltrated into a template, the accessibility of the glycan region may be further reduced compared to free soluble gp120 protein. Similarly, a reduction in epitope binding was also observed with 2G12 in MS nanocapsules.

Taken together, these results indicate that incorporation of protein into nanocapsules does reduce B-cell epitope availability somewhat. Of course, T-cell epitopes will not be similarly affected. Whether the gp120 nanocapsules can induce antibodies comparable to those assessed in this study will need further evaluation in animal trials.

To summarise, the two experiments detailed here demonstrate specific applications of two of the four nanocapsules that we have formulated. Further, we have previously demonstrated the utility of ovalbumin nanocapsules for delivering antigen to dendritic cells [12]. We predict that there may be several future applications of this platform technology.

## 5. Conclusions

This study demonstrates the formation of nanocapsules compromised of cross-linked protein only;Nanocapsules can be formed from a variety of proteins of differing masses and isoelectric points;Different-sized nanocapsules can be created.UreA nanocapsules were able to induce protective responses in a mouse model of *H. pylori* infection;gp120 nanocapsules retained the availability of B-cell epitopes;This technique may be a universally applicable platform for the formation or protein-only nanoparticles.

## Figures and Tables

**Figure 1 vaccines-12-00410-f001:**
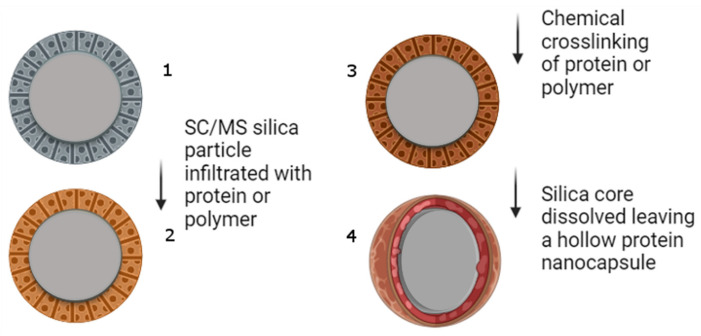
A representation of the process to produce protein nanocapsules. (1) Silica templates are constructed (only SC/MS shown here, but fully porous MS templates were also synthesised). The solid core with the surrounding porous shell is depicted. (2) Protein is incubated with the templates and infiltrates the pores. (3) Infiltrated protein is cross-linked, and in (4), the silica templates are dissolved leaving the protein capsule.

**Figure 2 vaccines-12-00410-f002:**
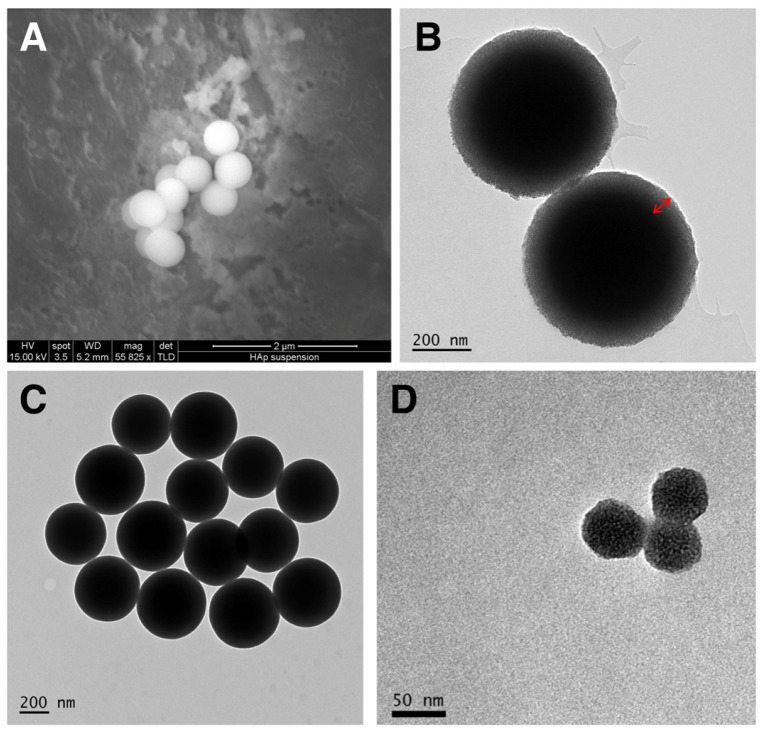
Electron micrographs of the silica templates synthesized. (**A**) SEM and (**B**) TEM of SC/MS templates. The red arrow indicates the mesoporous “shell” of the SC/MS template, into which protein is infiltrated. (**C**) TEM of the solid core particles used for the fabrication of SC/MS templates. (**D**) TEM of MS templates.

**Figure 3 vaccines-12-00410-f003:**
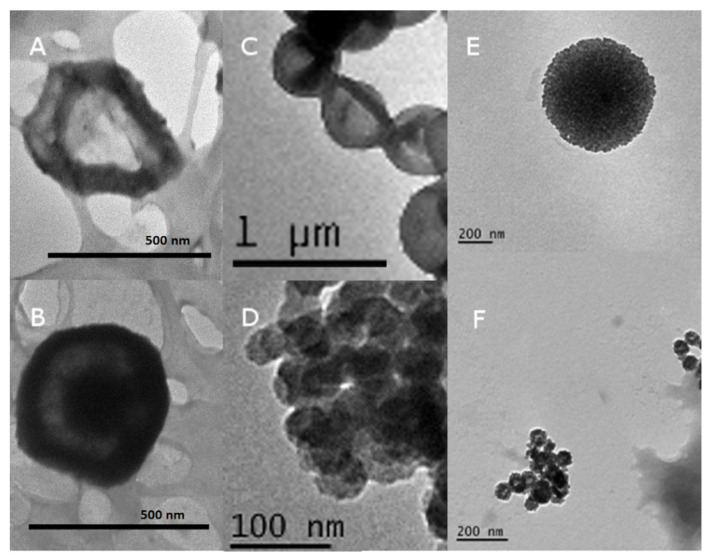
Representative TEM micrographs of synthesised nanocapsules. (**A**) Ovalbumin large, (**B**) C27 large, (**C**) UreA large, (**D**) UreA small, (**E**) gp120 large and (**F**) gp120 small.

**Figure 4 vaccines-12-00410-f004:**
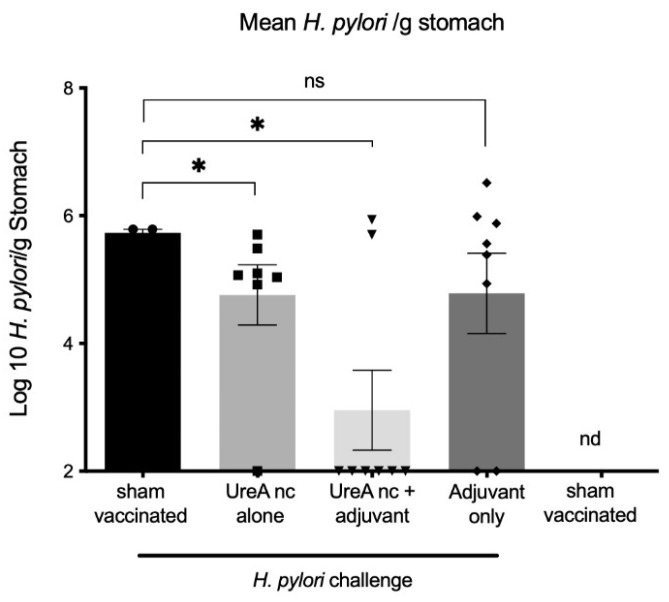
Detection of *H. pylori* burden by qPCR. Mice vaccinated with both UreA nanocapsules (nc), and UreA nc plus adjuvant had reduced colonization 21 days after challenge compared to controls (* *p* < 0.05). Mice that received adjuvant alone did not have significantly reduced levels (ns *p* > 0.05). *n* = 3–8 mice per group, SEM-indicated.

**Figure 5 vaccines-12-00410-f005:**
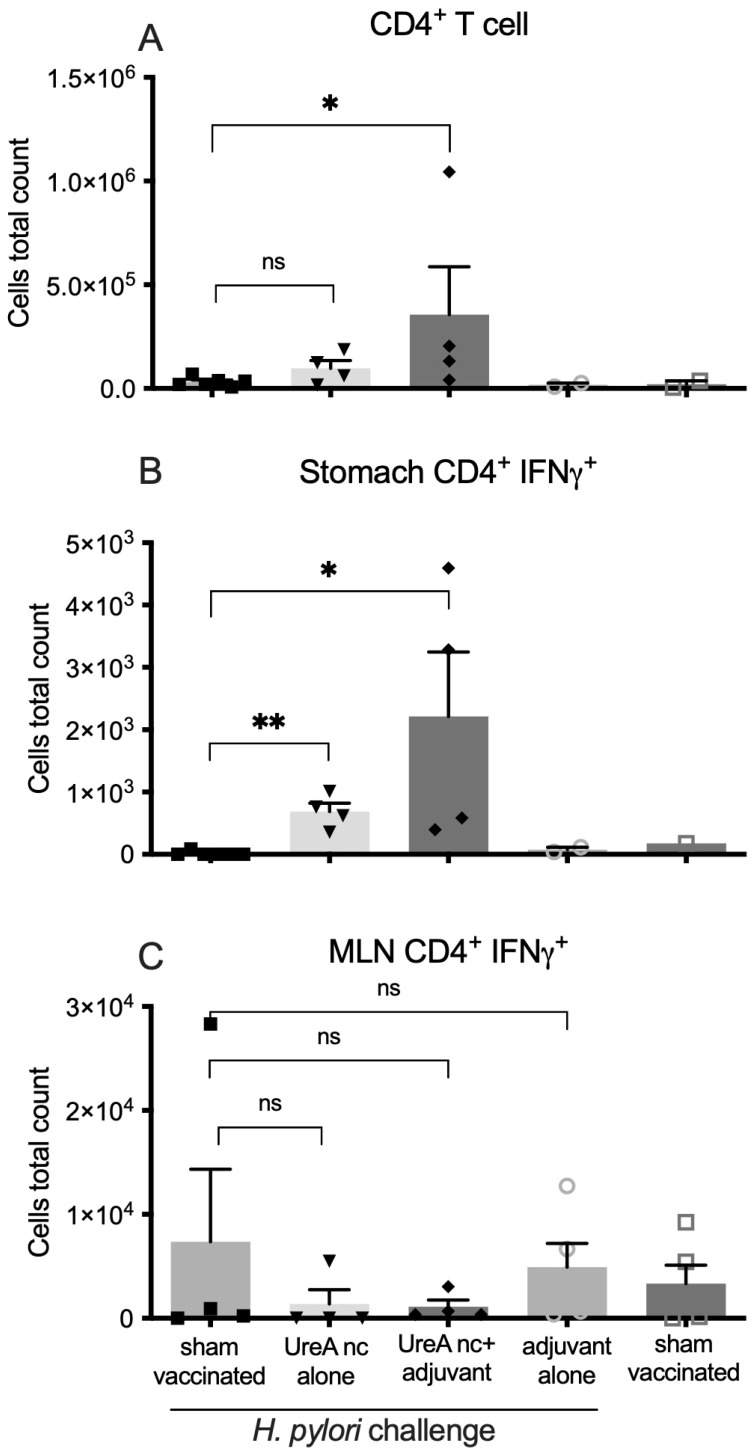
Characterization of lymphocyte populations in the stomach and mesenteric lymph nodes of vaccinated mice. Lymphocytes were isolated and analysed by flow cytometry. (**A**) CD4^+^ T cells in the stomach. (**B**) IFNγ-secreting CD4^+^ T cells in the stomach. (**C**) IFNγ-secreting CD4^+^ T cells in the MLN. Mice vaccinated with UreA nanocapsules or UreA nanocapsules with adjuvant had significantly increased infiltrates of CD4^+^ T cells and IFNγ-secreting CD4^+^ T cells in the stomach, but not the MLN. Bars represent mean and standard error. ns *p* > 0.05, * *p* < 0.05, ** *p* < 0.01. *n* = 3–5 mice per group, SEM-indicated.

**Figure 6 vaccines-12-00410-f006:**
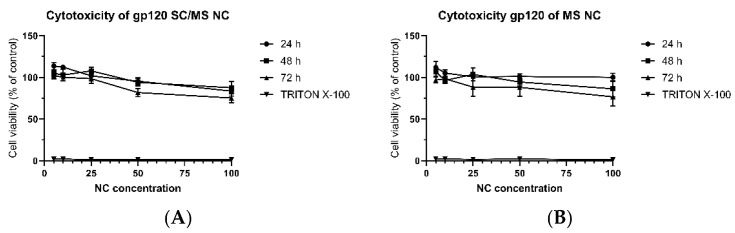
Cell viability of DC2.4 cells treated with (**A**) SC/MS gp120 and (**B**) MS gp120 nanocapsules at increasing concentrations (µg/mL). Cell viability is maintained above 70% for all concentrations across all time points, indicating limited toxic effects of gp120 nanocapsules to DC2.4 cells up to the highest concentrations of 100 µg/mL. *n* = 3, SEM-indicated. Cytotoxicity was found to be time-dependent (*p* = 0.0005) and dose-dependent (*p* ≤ 0.0001).

**Figure 7 vaccines-12-00410-f007:**
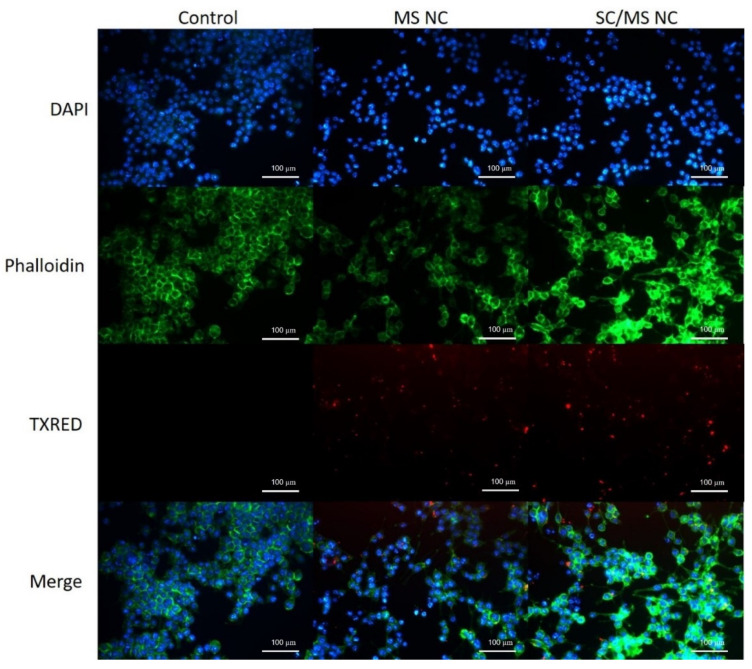
Fluorescence micrographs of DC2.4 cells with internalised gp120-based SC/MS and MS nanocapsules within 6 h. Nanoparticles can be seen internalised within the cell. Cell nuclei are shown in blue, the actin skeleton is shown in green, and fluorescent nanocapsules are shown in red.

**Figure 8 vaccines-12-00410-f008:**
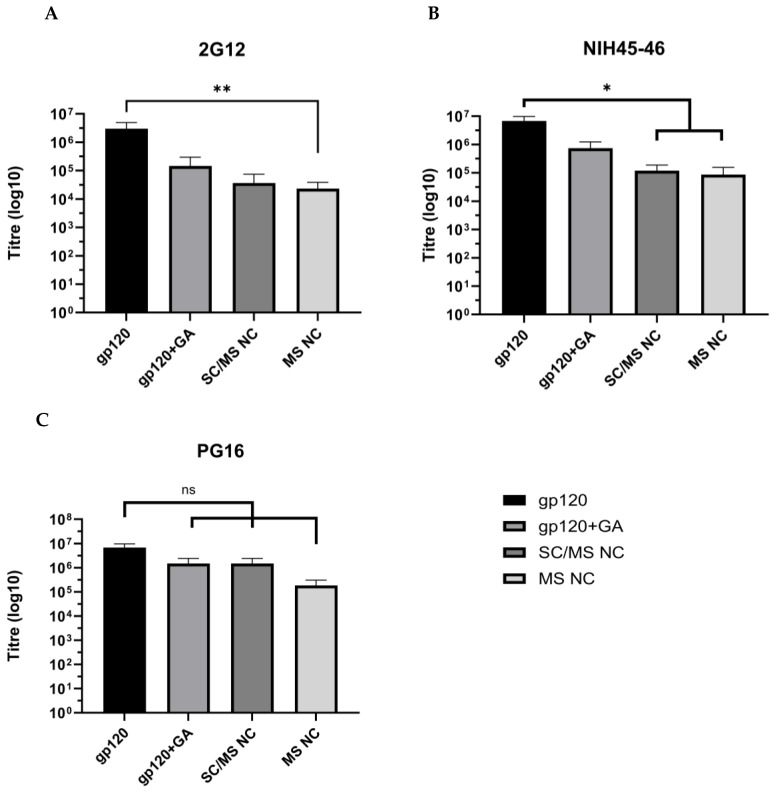
ELISA results showing antibody titres of HIV epitope-specific antibodies ((**A**–**C**), as labelled) in either soluble gp120 or nanocapsules. Titres across all three antibodies indicate reduced epitope availability in gp120 nanocapsules compared to soluble gp120, although this was not significant when probing with the PG16 antibody. *n* = 3, SEM indicated. ns *p* > 0.05, * *p* < 0.05, ** *p* < 0.01.

**Figure 9 vaccines-12-00410-f009:**
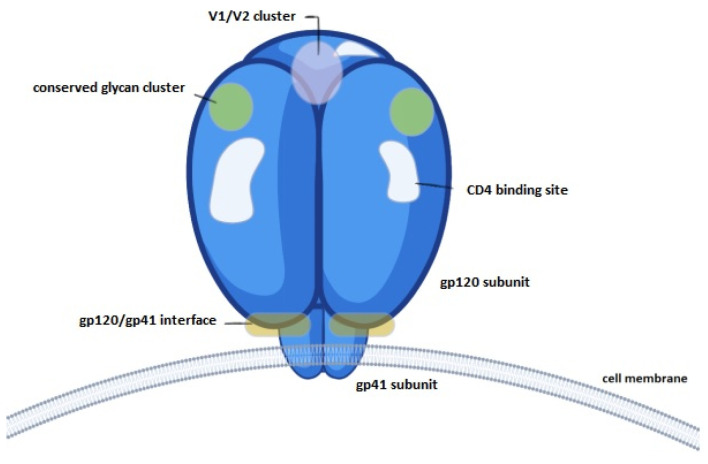
The HIV-1 gp160 trimer envelope spike protein consists of gp120 trimers in complex with gp41 timers. Key epitope regions highlight broadly neutralizing antibody targets against HIV-1; 2G12 targets the conserved glycan region, NIH45-46 targets the CD4 binding site and PG16 targets a conserved epitope in the V1/V2 loop domain.

**Table 1 vaccines-12-00410-t001:** Vaccination scheme and schedule.

Group	Vaccine Day 0 and Boost Day 14	Dose per Vaccination	Challenged (Day 35)
Sham-vaccinated	PBS	50 µL PBS	Yes
UreA nanocapsules	Nanocapsules without adjuvant	10 µg of nanocapsule in 50 µL PBS	Yes
UreA nanocapsules + adjuvant	Nanocapsules with TiterMax^®^ Gold adjuvant	10 µg of nanocapsule in 50 µL TiterMax/PBS	Yes
Adjuvant only	TiterMax^®^ Gold adjuvant only	50 µL TiterMax/PBS	Yes
Sham-vaccinated, no challenge	PBS	50 µL PBS	No

**Table 2 vaccines-12-00410-t002:** *H. pylori* 16S primers for qPCR amplification [18].

Forward	5′-CTTAACCATAGAACTGCATTTGAAACTAC-3′
Reverse	5′-GGTCGCCTTCGCAATGAGTA-3′

**Table 3 vaccines-12-00410-t003:** Properties of the constructed nanocapsules.

Protein	MW (Including Tag)	pI (Including Tag)	TEM Diameter	Hydrodynamic Size	Zeta Potential
Ovalbumin	44.5 (native)	4.5	516 ± 20 nm (SC/MS) 41 ± 2.5 nm (MS)	410 nm (SC/MS) 48 nm (MS)	Not determined
C27	13.6 kDa	6.81	Approx. 500 nm (SC/MS)	Not determined	Not determined
Urease A	28 kDa	7.4	523.3 ± 13.62 nm (SC/MS) 40 ± 9.21 nm (MS)	543 nm (SC/MS) 51nm (MS)	−17 mV (SC/MS) −14 mV (MS)
gp120	120 kDa	4.97	520.2 ± 15.32 nm (SC/MS) 56.6 ± 6.17 nm (MS)	487 nm (SC/MS) 56 nm (MS)	−15 mV (SC/MS) −14 mV (MS)

## Data Availability

Data are contained within the article. We encourage contacting the authors for any further explanation required.

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
