# Peer review of "Nanocapsules Comprised of Purified Protein: Construction and Applications in Vaccine Research"

_vaccines, 2024, doi:10.3390/vaccines12040410_

Round 1

Reviewer 1 Report

Comments and Suggestions for Authors

This manuscript detailing the preparation of silica nanoparticles and subsequent templating of proteins to create protein nanoparticles represents a work at the forefront of the promising field of encapsulation and delivery of therapeutics. The scientific approach with respect to the in vitro studies appears solid and the results showing the formation of the nanoparticles reveal meticulous work.

-the introduction does not adequately describe the novelty of this work or how it relates to or advances on previous work in the field. In fact, details about what this work actually entails are mostly absent from this rather vague portion. The introduction should frame the work and demonstrate that the authors are aware of how it fits in the extremely dense field it occupies.

-It is actually the abstract that describes the work done most clearly. If this is a stylistic element of this journal then this comment can be ignored. 

-The intro stresses the importance of mrna delivery but doesn't describe in detail any of the systems in existence that do this.

-The introduction could also establish more clearly that several different particle types were prepared but not used for any applications in the study. Perhaps this is a discussion for the editor but it seems like a confusing choice to prepare the ovalbumin and c27 particles just to show their  size. If this work is about the potential use for the non urea and gp120 types, shouldn't something actually be done with those? Otherwise, it could be argued that this work is sufficient to just show the preparation and application of the urea and gp120 types. 

-The methods mention 80 volume% of 2 m hf. What is the other component? If it is just water, this description is needlessly ambiguous. Directly stating the actual hf molarity would be much clearer and appropriate. If it is diluted into ammonium fluoride at some concentration, this should be mentioned. 

-The manuscript uses the language of encapsulation, calling the particles nanocapsules throughout. It should be clearly described that no cargo is actually loaded or released. From the methods and discussion, it appears that the capsule shell itself is the active agent. Is this correct? This is a somewhat confusing aspect of the study and could be more clearly explained.

-The methods mention that the vaccine trial is modeled from a previous study. Why not just provide a short but informative description of what was done in the current work? Is there a page limit for this journal? What modifications were used? Just describing the methods properly would make more sense and eliminate this unnecessary obfuscation of detail.

-I am not sure if this study really needs the ovalbumin and c27 particles as they aren't used for anything meaningful. I don't think it would constitute an unnecessary fragmentation of the work to split those off into a separate study since they don't contribute to any of the meaningful results this study seems to be focused on. in fact, they are more of a distraction since the introduction is so sparse and vague; the effect is that the reader is left wondering if their application will be demonstrated anywhere in the work, only to find out these particles were made just for demonstration. 

-the cell and biological methods seem solid but the writing and explanation about the nanoparticle systems themselves are shallow and completely lacking in clarity. the overall effect is a work in which the primary focus is unclear. is this a proof of concept that nanoparticles can be made or a demonstration of their utility? both at once doesn't feel like it works that well. the results for gp120 and ureA look solid. why detract from these by showing ovalbumin, a relatively routine system, and c27, something interesting but mysteriously not utilized in the work, without any application?

Author Response

This manuscript detailing the preparation of silica nanoparticles and subsequent templating of proteins to create protein nanoparticles represents a work at the forefront of the promising field of encapsulation and delivery of therapeutics. The scientific approach with respect to the in vitro studies appears solid and the results showing the formation of the nanoparticles reveal meticulous work.

-the introduction does not adequately describe the novelty of this work or how it relates to or advances on previous work in the field. In fact, details about what this work actually entails are mostly absent from this rather vague portion. The introduction should frame the work and demonstrate that the authors are aware of how it fits in the extremely dense field it occupies.

RESPONSE. That is a good point. The introduction has been expanded, with more background on the nanocapsules (that were originally designed for drug delivery). In particular, the following was added to frame the background on the idea of the nanocapsules:

“The rationale for this work was to use the “templating” approach first described as a vehicle for the delivery of a cargo in drug delivery (Wang et al., 2008). They were the first to use solid core/mesoporous shell (SC/MS) silica particle templates in this way, using a variety of macromolecules to form the capsules. We reasoned that by using protein as the macromolecule we could make a potentially immunogenic particle, of a defined, monodispersed size (as the size is dependent on that of the template). These are not designed to deliver a cargo (although potentially could).”

For the reviewer’s information, the reference to be inserted is Wang et al., 2008. Nano. Letters, 8(6):1741-5. doi: 10.1021/nl080877c.

-It is actually the abstract that describes the work done most clearly. If this is a stylistic element of this journal then this comment can be ignored. 

RESPONSE. Good comment- an abstract is clearly very important.

-The intro stresses the importance of mrna delivery but doesn't describe in detail any of the systems in existence that do this.

RESPONSE. Indeed, this needed references, which have been added. Text has been slightly modified, as follows:

“No introduction is complete without mention of the nanoparticles used to deliver mRNA, including but not limited to lipid nanoparticles (refs inserted).”

For the reviewer’s information, the inserted references are Zhang et al. 2024. ACS Omega 9(6):6219-6234. doi: 10.1021/acsomega.3c08353.

Parhiz et al., 2024. Lancet. 7:S0140-6736(23)02444-3. doi: 10.1016/S0140-6736(23)02444-3.

-The introduction could also establish more clearly that several different particle types were prepared but not used for any applications in the study. Perhaps this is a discussion for the editor but it seems like a confusing choice to prepare the ovalbumin and c27 particles just to show their  size. If this work is about the potential use for the non urea and gp120 types, shouldn't something actually be done with those? Otherwise, it could be argued that this work is sufficient to just show the preparation and application of the urea and gp120 types. 

RESPONSE. This is a good point, and indeed one that we had been considering ourselves. When we proposed the manuscript idea to the editor, we highlighted it as something between a methods paper and a paper giving strong research results. The plan was to demonstrate the utility of nanocapsules as a platform technology. We had previously published the ovalbumin nanocapsules (Taki et al 2020). The C27 capsules were made as we wanted to see if we could make stable nanocapsules from a protein of relatively low molecular mass. They were not characterised further. We have amended the text in the introduction to make this clear. In particular, the following was added.

“Lastly, we evaluated two of these in experimental systems. It should be noted that of the four nanocapsules described here, two are for illustrative purposes and two for experimental analysis. The aim was to demonstrate that nanocapsules can be synthesised from a range of proteins. Ovalbumin nanocapsules have been described by us previously (Taki et al 2020), while C27 nanocapsules were constructed as we wanted to see if we could make stable nanocapsules from a protein of relatively low molecular mass. This was achieved. They were not characterised further. Nanocapsules constructed from urease A (UreA) from Helicobacter pylori were tested in a mouse model of infection, while those constructed from the gp120 protein from the HIV virus were tested for immunogenicity in vitro.

If necessary, we could remove either one or both of Ova and C27, but we do think that leaving them in adds, rather than detracts from the paper, demonstrating that a range of different proteins can be used to formulate nanocapsules.

-The methods mention 80 volume% of 2 m hf. What is the other component? If it is just water, this description is needlessly ambiguous. Directly stating the actual hf molarity would be much clearer and appropriate. If it is diluted into ammonium fluoride at some concentration, this should be mentioned. 

RESPONSE. Good point. Amended. It is 2 M HF treatment in an 8 M ammonium fluoride buffer (pH 5).

-The manuscript uses the language of encapsulation, calling the particles nanocapsules throughout. It should be clearly described that no cargo is actually loaded or released. From the methods and discussion, it appears that the capsule shell itself is the active agent. Is this correct? This is a somewhat confusing aspect of the study and could be more clearly explained.

RESPONSE. Yes, this is correct and a good comment. This relates to your comment about the introduction, and as mentioned in response to your first point, text has been added addressing this.

-The methods mention that the vaccine trial is modeled from a previous study. Why not just provide a short but informative description of what was done in the current work? Is there a page limit for this journal? What modifications were used? Just describing the methods properly would make more sense and eliminate this unnecessary obfuscation of detail.

RESPONSE. We have had discussion with the journal regarding the methods. They advised they can be left as they are. We do think that as described, and with the references, the methods are sufficient and informative.

-I am not sure if this study really needs the ovalbumin and c27 particles as they aren't used for anything meaningful. I don't think it would constitute an unnecessary fragmentation of the work to split those off into a separate study since they don't contribute to any of the meaningful results this study seems to be focused on. in fact, they are more of a distraction since the introduction is so sparse and vague; the effect is that the reader is left wondering if their application will be demonstrated anywhere in the work, only to find out these particles were made just for demonstration. 

RESPONSE. Agree with this comment that it needs to be better explained. Please see the reworked introduction as detailed in your points above. Your comments have been useful and have hopefully made it clearer in the re-write.

-the cell and biological methods seem solid but the writing and explanation about the nanoparticle systems themselves are shallow and completely lacking in clarity. the overall effect is a work in which the primary focus is unclear. is this a proof of concept that nanoparticles can be made or a demonstration of their utility? both at once doesn't feel like it works that well. the results for gp120 and ureA look solid. why detract from these by showing ovalbumin, a relatively routine system, and c27, something interesting but mysteriously not utilized in the work, without any application?

RESPONSE. As per above, we think the re-worked introduction sets the scene better.

Reviewer 2 Report

Comments and Suggestions for Authors

The authors report on formation and characterization of protein nanocapsules built on silica templates. However, there is no clear formulated advantage of the formed capsules with respect to protein (nano)particles. Capsules are supposed to serve for encapsulation of certain active compounds that is not the case here. The applied procedure for the removal of silica template (using concentrated HF) is very harsh and may significantly chemically modify the used proteins. The functionality of the proteins forming the capsules may be changed and the risk of triggering an uncontrolled immune response may be high for fluorinated proteins. The safety issues should be addressed properly and the actual composition of the nanocapsules (e.g. content of fluorine, surfactants) should be determined. This is critical as failure in showing native protein-based capsules may question the whole procedure requiring such harsh conditions for the removal of the template. By the way other templates like calcium carbonate might be used that require much milder conditions for its removal.

Figure 7. What is the source of fluorescence of crosslinked nanocapsules (red fluorescence). The capsules alone should be studied to show their fluorescence. Rather higher magnifications should be used to see undoubtedly the nanocapsules within the cells.

The results on H. pylori colonisation are not very convincing - the data are very scattered and the effect is rather weak (1 log reduction in colonization only). Above all, the studied Urease A capsules have been already reported by the authors (reference 6) so the novelty of the current report is limited.

Author Response

The authors report on formation and characterization of protein nanocapsules built on silica templates. However, there is no clear formulated advantage of the formed capsules with respect to protein (nano)particles. Capsules are supposed to serve for encapsulation of certain active compounds that is not the case here.

RESPONSE. In response to this and similar comments from other reviewers we have reworded the introduction to hopefully make it clearer. One major advantage of using a template is that the size and homogeneity of the particles is easily controlled. In particular, the following has been added:

“The rationale for this work was to use the “templating” approach first described as a vehicle for the delivery of a cargo in drug delivery (Wang et al., 2008). They were the first to use solid core/mesoporous shell (SC/MS) silica particle templates in this way, using a variety of macromolecules to form the capsules. We reasoned that by using protein as the macromolecule we could make a potentially immunogenic particle, of a defined, monodispersed size (as the size is dependent on that of the template). These are not designed to deliver a cargo (although potentially could).”

For the reviewer’s information, the reference to be inserted is Wang et al., 2008. Nano. Letters, 8(6):1741-5. doi: 10.1021/nl080877c.

The applied procedure for the removal of silica template (using concentrated HF) is very harsh and may significantly chemically modify the used proteins. The functionality of the proteins forming the capsules may be changed and the risk of triggering an uncontrolled immune response may be high for fluorinated proteins. The safety issues should be addressed properly and the actual composition of the nanocapsules (e.g. content of fluorine, surfactants) should be determined. This is critical as failure in showing native protein-based capsules may question the whole procedure requiring such harsh conditions for the removal of the template. By the way other templates like calcium carbonate might be used that require much milder conditions for its removal.

RESPONSE. HF has long been used in protein chemistry (proteins are compatible with and highly soluble in HF) and a recent paper has shown that fluorination of a protein had no significant impact on stability or folding kinetics of the protein (Welte et al., 2020 Scientific Reports volume 10, Article number: 2640). The findings that the nanocapsules exhibit relatively low toxicity (Figure 6) and are taken up by live cells (Figure 7) do argue that they are bio-compatible.

Figure 7. What is the source of fluorescence of crosslinked nanocapsules (red fluorescence). The capsules alone should be studied to show their fluorescence.

RESPONSE. The autofluorescence is due to the cross-linking agent, glutaraldehyde. This is a well-known phenomenon, and indeed, can be a hindrance in tissues fixed with it, precluding fluorescent labelling. However, in our application it is an advantage. All the capsules contain protein cross-linked with glutaraldehyde, but we didn’t study them alone. Unsure what the advantage would be.

Rather higher magnifications should be used to see undoubtedly the nanocapsules within the cells.

RESPONSE. Unfortunately, these are the only micrographs available. We do believe they are indicative of uptake.

The results on H. pylori colonisation are not very convincing - the data are very scattered and the effect is rather weak (1 log reduction in colonization only). Above all, the studied Urease A capsules have been already reported by the authors (reference 6) so the novelty of the current report is limited.

RESPONSE. Yes, we acknowledge that the protection is not all that high- certainly not sterilizing immunity! This is a limitation of this mouse model (and as an aside, perhaps why there is as yet no vaccine against H. pylori). The reason for doing two trials was that we wanted to confirm that the protection was reproducible.

Reviewer 3 Report

Comments and Suggestions for Authors

The manuscript is devoted to the vaccine immunogen construction, a topic that is a hot spot in rapid designing vaccines against newly emerging infections. Hence this manuscript totally corresponds to the journal subject content and may be of a high interest to the readers. 

However, certain notes are to be made highlighting some drawbacks  that should be taken into attention in order to improve the manuscript. 

1) The previous paper from the same group (ref. 6) describes the data concerning the immunogenicity of nanocapsules made from H. pylori urease in a more profound manner than in the presented manuscript. Are the presented data (which comprise the probable self-plagiarism) the same that have been obtained earlier or obtained in a separate experiment? In the latter case, what was the reason of repeating the experiment? 

2) Why different protein nanocapsules are subject to different assays? E.g., toxicity has been studied for nanocapsules made of gp120, and not for other ones; nanocapsules made of hen egg albumin and influenza hemagglutinin fragment have been only partially studied compared to others. Hen egg albumin nanocapsules seem not to be so interesting as those corresponding to infectious immunogens and can be excluded from the consideration.

3) Electron micrographs of mesoporous particles show their aggregation. In this case the hydrodynamic size estimation is of a great interest. However, these data are not present, and one cannot see the size distribution of MS particles in their solutions. These data should be presented, at least as a Supplementary material.   

4) Use of concentrated HF for silica gel dissolution may cause damage for proteins that form nanocapsules, e.g. partial hydrolysis of amide bonds and removal of some protein fragments from the nanocapsule cores. The authors do not confirm the protein integrity, however it should be done at least via confirming the absence of peptide material in washings after the silica gel dissolutions. removal of certain fragments via partial protein hydrolysis may be one of the reasons of the alterations in antibody binding to protein nanocapsules. 

5) Despite of the authors' opinion, alterations in antigen specificities in the proteins in nanocapsules should be considered significant, taking into account about 100-fold or more changes in antibody binding.  

6) It should be emphasized that protein nanocapsules are almost not immunogenic in the absence of an adjuvant.

Author Response

The manuscript is devoted to the vaccine immunogen construction, a topic that is a hot spot in rapid designing vaccines against newly emerging infections. Hence this manuscript totally corresponds to the journal subject content and may be of a high interest to the readers. 

RESPONSE. Thank you. We hope so!

However, certain notes are to be made highlighting some drawbacks  that should be taken into attention in order to improve the manuscript. 

  • The previous paper from the same group (ref. 6) describes the data concerning the immunogenicity of nanocapsules made from H. pylori urease in a more profound manner than in the presented manuscript. Are the presented data (which comprise the probable self-plagiarism) the same that have been obtained earlier or obtained in a separate experiment? In the latter case, what was the reason of repeating the experiment? 

RESPONSE. This was a second trial as we wanted to confirm that the protection was reproducible. The data in the previous manuscript and this one is completely separate. However, methods were almost identical, which is why the “plagiarism” exits. We have explained to the editors that this is the case, and they have advised that it is OK.

  • Why different protein nanocapsules are subject to different assays? E.g., toxicity has been studied for nanocapsules made of gp120, and not for other ones; nanocapsules made of hen egg albumin and influenza hemagglutinin fragment have been only partially studied compared to others. Hen egg albumin nanocapsules seem not to be so interesting as those corresponding to infectious immunogens and can be excluded from the consideration.

RESPONSE. This is a good point, and one made by another reviewer so we will repeat the explanation. When we proposed the manuscript idea to the editor, we highlighted it as something between a methods paper and a paper giving strong research results. The plan was to demonstrate the utility of nanocapsules as a platform technology. We had previously published the ovalbumin nanocapsules (Taki et al 2020). The C27 capsules were made as we wanted to see if we could make stable nanocapsules from a protein of relatively low molecular mass. They were not characterised further. We have amended the text to make this clear. Added text as follows:

“It should be noted that of the four nanocapsules described here, two are for illustrative purposes and two for experimental analysis. The aim was to demonstrate that nanocapsules can be synthesised from a range of proteins. Ovalbumin nanocapsules have been described by us previously (Taki et al 2020), while C27 nanocapsules were constructed as we wanted to see if we could make stable nanocapsules from a protein of relatively low molecular mass. This was achieved. They were not characterised further. Nanocapsules constructed from urease A (UreA) from Helicobacter pylori were tested in a mouse model of infection, while those constructed from the gp120 protein from the HIV virus were tested for immunogenicity in vitro.”

Regarding the ovalbumin capsules, we will be guided by the Editor whether to include or omit it.

  • Electron micrographs of mesoporous particles show their aggregation. In this case the hydrodynamic size estimation is of a great interest. However, these data are not present, and one cannot see the size distribution of MS particles in their solutions. These data should be presented, at least as a Supplementary material.   

RESPONSE. We do not have the individual distribution graphs. However, the close matching of the hydrodynamic size with the TEM diameter would indicate minimal aggregation. In our experience, some aggregation using TEM is inevitable.

  • Use of concentrated HF for silica gel dissolution may cause damage for proteins that form nanocapsules, e.g. partial hydrolysis of amide bonds and removal of some protein fragments from the nanocapsule cores. The authors do not confirm the protein integrity, however it should be done at least via confirming the absence of peptide material in washings after the silica gel dissolutions. removal of certain fragments via partial protein hydrolysis may be one of the reasons of the alterations in antibody binding to protein nanocapsules. 

RESPONSE. That it an interesting point, again made by another reviewer. HF has long been used in protein chemistry (proteins are compatible with and highly soluble in HF). In terms of integrity, we did not look for peptides in washings after template removal. That is indeed something that we may be able to do in the future. We do think that at least by TEM the nanocapsules are intact, however of course that doesn’t mean some protein material may have been removed. In the end, that may not matter, as it would be unlikely that a specific part of each protein would be removed. Perhaps Mass Spectrometry analysis could answer this, but it is beyond the scope of this investigation.

  • Despite of the authors' opinion, alterations in antigen specificities in the proteins in nanocapsules should be considered significant, taking into account about 100-fold or more changes in antibody binding.  

RESPONSE. This is a good point and have substituted the word “significant” for “somewhat” and amended the discussion as follows:

“The results indicated that in two of three epitopes, availability was significantly reduced in both SC/MS and MS nanocapsules compared to soluble gp120. Availability of the PG16 epitope was not significantly reduced. The reduction in availability is not unexpected, as some B cell epitopes will no longer be on the surface, thereby reducing availability per unit mass of nanocapsule compared to free protein.”

6) It should be emphasized that protein nanocapsules are almost not immunogenic in the absence of an adjuvant.

RESPONSE. Good point. Have added a statement as follows, addressing this.

“The mean number of cells was higher in the nanocapsules plus adjuvant group, indicating that the adjuvant may be required for maximal recruitment of cells.”

Reviewer 4 Report

Comments and Suggestions for Authors

Overview and general recommendation:

In the research, the authors create nanocapsules in different size (500nm and 50nm) and characterize the nanocapsules from four different proteins. They establish that nanocapsules from urease A subunit of H. pylori can reduce infection in vaccinated mouse and nanocapsules  from HIV gp120 protein can be up-taken by cells and can be recognized by antibody.

I find the paper is well-organized. Major methods are well described in the manuscript and properly used in the research. The results are strong enough to support their conclusions. However, I still think there is something to be improved. I suggest the authors use more methods to evaluate the immune response of cells or mouse after vaccinated with nanocapsules from different proteins.

Major comments:

1.      I think various methods should be included to describe the immune response of cells or mouse after vaccinated with nanocapsules from urease A subunit of H. pylori or HIV gp120. For example, they can measure the expression of some immune factors.

2.      Can the authors add some discussion about the advantages and disadvantages of nanocapsules compared with other methods?

Author Response

In the research, the authors create nanocapsules in different size (500nm and 50nm) and characterize the nanocapsules from four different proteins. They establish that nanocapsules from urease A subunit of H. pylori can reduce infection in vaccinated mouse and nanocapsules  from HIV gp120 protein can be up-taken by cells and can be recognized by antibody.

I find the paper is well-organized. Major methods are well described in the manuscript and properly used in the research. The results are strong enough to support their conclusions. However, I still think there is something to be improved. I suggest the authors use more methods to evaluate the immune response of cells or mouse after vaccinated with nanocapsules from different proteins.

RESPONSE. That’s a good comment. We had done this in a previous study (Skakic et al., 2023) but sera collection before challenge was not done here.

Major comments:

  1. I think various methods should be included to describe the immune response of cells or mouse after vaccinated with nanocapsules from urease A subunit of H. pylori or HIV gp120. For example, they can measure the expression of some immune factors.

RESPONSE. See the response above. Regarding gp120 nanocapsules, vaccinating mice is indeed something that should be done in the future but was beyond the scope of the current study.

  1. Can the authors add some discussion about the advantages and disadvantages of nanocapsules compared with other methods?

RESPONSE. We have extensively modified the introduction to give more background on the nanocapsules and talk about the advantages and disadvantages.

Round 2

Reviewer 1 Report

Comments and Suggestions for Authors

All the comments have been adequately addressed and the authors have made convincing arguments for the inclusion of the illustrative ovalbumin and c27 systems. The results were always solid so these changes set the manuscript up for success.

Author Response

We thank the reviewer for the comments. Also for the initial review, as it has contributed greatly to enhancing the manuscript.

Reviewer 2 Report

Comments and Suggestions for Authors

The authors responded to my concerns but their are not satisfactory. My general impression is still negative mainly due to the toxicity problems related to using HF which reacting with silica leads not only to UNCONTROLLED
fluorination of proteins but also produces water soluble H2SiF6 that the authors do not address and do not remove (control of Si and F content in the dispersion of the capsules would be necessary).

The cited Welte et al (Sci Rep, 2020) refers to labeling of selected aminoacids with SINGLE fluorine atoms while the authors here do not control the fluorination process.

I cannot agree with the advantage of formulation the capsules that is their controlled size as the authors do not show any importance of this size and above all do not intend to form the capsules to carry cargo molecules. Due tot the mentioned reasons I cannot recommend  publication of this report and suggest the authors at least to change the template to e.g. calcium carbonate that can be released in mild conditions using e.g. EDTA.

Author Response

The authors responded to my concerns but their are not satisfactory. My general impression is still negative mainly due to the toxicity problems related to using HF which reacting with silica leads not only to UNCONTROLLED
fluorination of proteins but also produces water soluble H2SiF6 that the authors do not address and do not remove (control of Si and F content in the dispersion of the capsules would be necessary).

RESPONSE. HF is known to be compatible with protein, and indeed, protein is highly soluble in HF. After the dissolution of the silica templates, the capsules are extensively washed to ensure the removal of HF. Further, the in vitro toxicity studies we have undertaken reveal some, but not remarkable, toxicity at high concentrations of nanocapsules. This is not uncommon and is a well known feature of many nanoparticles, as we mention in the manuscript. Lastly, no adverse effects on mice were found after administration of the capsules, indicating that they were not toxic. We have shown this over several trials. 

The cited Welte et al (Sci Rep, 2020) refers to labeling of selected aminoacids with SINGLE fluorine atoms while the authors here do not control the fluorination process.

That is a fair point, yes that is what they were attempting to do. The point we are making here is that it does not modify the integrity of the protein. We agree that we do not test for multiple fluorination, however given the points above we do not feel it necessary to do so. However, we will certainly keep this in mind and may do so in future experimentation.

I cannot agree with the advantage of formulation the capsules that is their controlled size as the authors do not show any importance of this size and above all do not intend to form the capsules to carry cargo molecules. Due tot the mentioned reasons I cannot recommend  publication of this report and suggest the authors at least to change the template to e.g. calcium carbonate that can be released in mild conditions using e.g. EDTA.

RESPONSE. Regarding the importance of size, it is well known that the different sizes of nanoparticles can be taken up in different ways, and this influences the resulting immune response. This is commented on and referenced in the manuscript (ref 26).

In regard to the second point, the use of calcium carbonate, that is a good point, however we cannot be expected to repeat our experiments using a different system. We will, however, consider this for the future and indeed perhaps a comparison between the systems can be made. 

Reviewer 3 Report

Comments and Suggestions for Authors

I agree with all authors' answers to my notes except the absence of graphs of nanoparticle size distribution obtained by hydrodynamic size measurements. These graphs can clearly demonstrate the uniformity of nanoparticle sizes in a solution and absence of their aggregation. Without these data a reader can doubt about it. Besides that, there is almost no description of the method of the hydrodynamic size measurement.   

Author Response

I agree with all authors' answers to my notes except the absence of graphs of nanoparticle size distribution obtained by hydrodynamic size measurements. These graphs can clearly demonstrate the uniformity of nanoparticle sizes in a solution and absence of their aggregation. Without these data a reader can doubt about it. Besides that, there is almost no description of the method of the hydrodynamic size measurement.   

RESPONSE. The point of the reviewer is taken. However, from the zetasizer we take the two measurements, of average size and zeta-potential. Over Easter we went back to the laboratory and looked at the files on the machine, however, the files for gp120 were not available. This manuscript is describing work done over several years, and as such the gp120 files have been overwritten by newer data. In terms of the sizing, the TEM analysis quite clearly shows the diameter of the capsules. I understand the reviewers comment about the size range and aggregation. Howerer the sizes calculated by TEM and the zetasizer are in accordance.

In terms of the description of the method of determination, this is an extremely common and standard technique, hence we did not deem it necessary to add details. We could add some text detailing the machine used and concentrations of nanocapsules measured, as we have done for TEM. Would that be desirable?

Reviewer 4 Report

Comments and Suggestions for Authors

I find the paper largely improved, and I have no doubt about recommending it for publication.

Author Response

(The authors gave the same response as above.)

Round 3

Reviewer 2 Report

Comments and Suggestions for Authors

I sustain my decision due to unresolvable methodological problems.

Reviewer 3 Report

Comments and Suggestions for Authors

The question of the absence of protein nanoparticle aggregation still remains unanswered.  The state of the nanoparticle preparations can define their immunogenic properties, that's why it should be properly studied. 

The type of the hydrodynamic size determination procedure, the equipment used and test conditions should be defined anyway in Materials and Methods section.